# OpenReview forum: "CURATe: Benchmarking Personalised Alignment of Conversational AI Assistants"
_NeurIPS.cc/2025/Datasets_and_Benchmarks_Track — Submitted to NeurIPS 2025 Datasets and Benchmarks Track_

### Official Review · Reviewer_Cb54 · 2025-06-11

**Rating:** 4
**Confidence:** 3

**Summary:**

The author introduce a benchmark evaluating LLM assistants’ handling of safety-critical user contexts, revealing consistent failures in personalization and safety. Explicit safety prompts improve results, highlighting the need for better self-reflection, user modeling, and risk assessment in AI assistants.

**Dataset Code Accessibility:**

Yes

**Ethical Considerations:**

No, there are no or only very minor ethics concerns

**Final Justification:**

The authors have addressed my comments. I maintain my positive score. Thank you.

**Limitations Weaknesses:**

1. While the benchmark covers five scenarios with 337 use cases each, it would be helpful to justify whether this scale is sufficient for comprehensive evaluation. Can the authors provide comparisons to prior benchmarks to support the dataset’s adequacy?

2. The paper could be further strengthened by exploring how the benchmark relates to concrete personalized AI assistant applications—such as psychological counseling [1], smartphone agents [2], housekeeping robots [3], and medical assistants [4]. Discussing how the benchmark offers insights into these domains would enhance its practical relevance. Additionally, more discussion of existing benchmarks and surveys on personalized LLM-based agents [5][6] would provide better context and highlight this work's unique contributions.

[1] Memorybank: Enhancing large language models with long-term memory. AAAI 2024

[2] Crafting personalized agents through retrieval-augmented generation on editable memory graphs. EMNLP 2024

[3] LLM-Personalize: Aligning LLM Planners with Human Preferences via Reinforced Self-Training for Housekeeping Robots. COLING 2025

[4] Memory-augmented llm personalization with short-and long-term memory coordination. NAACL 2024

[5] Do LLMs Recognize Your Preferences? Evaluating Personalized Preference Following in LLMs. ICLR 2025

[6] Personal LLM Agents: Insights and Survey about the Capability, Efficiency and Security. arXiv 2024

3. The current presentation is somewhat difficult to follow. Including an overall system or evaluation pipeline diagram could improve clarity and help readers better grasp the benchmark’s design and contributions.

**Strengths Contributions:**

1. It introduces a multi-turn benchmark to evaluate personalized alignment in LLM-based AI assistants, focusing on safety-critical user contexts.

2. Testing ten top models reveals common failures like ignoring conflicting preferences, sycophancy, and inconsistent use of user-specific information.

3. Prompting models with explicit safety context improves results, highlighting the need for self-reflection, user modeling, and dynamic risk assessment in future AI assistants.

---

> ### Author Response · Authors · 2025-08-01
> **Responses to suggestions - application domain is personal agentic AI assistants**
>
> We thank the reviewer for their suggestions and consideration of the strengths of our paper. We address their comments and concerns in turn below.
>
> "comparisons to prior benchmarks to support the dataset’s adequacy"
> The amount of conversations we tested in each scenario is not arbitrarily based on the number of entries of prior benchmarks, but on the specific problem we aimed to test for, which was quite novel. Due to the technical complexity of our benchmark, i.e. testing for user-specific safety risks across four categories (common allergies, phobias, etc.) in ways that are realistic (e.g. in likely model recommendations), we were necessarily restricted. The process of arriving at the exact number of entries is described in Appendix B. The reason was that we wanted to balance having enough entries to be significant while maintaining quality such that scenarios do not become repetitive or too convoluted. We also wanted to balance the number of conversations per risk category to make an unbiased assessment, and some categories were harder to create high-quality examples (e.g. common allergies and trauma triggers) than others (e.g. common phobias), especially ones that a model would recommend an activity for in a realistic conversation. We arrived at this number as after this point entries started becoming repetitive and hence less useful.
>
> "exploring how the benchmark relates to concrete personalized AI assistant applications"
>
> The domain of interest for us is personal agentic AI assistants, which is a specific application distinct from the ones the reviewer suggested. These are commonly discussed as the next frontier of AI assistants, blending the function of personal assistants like Siri or Alexa (e.g. answering queries) with the dialogical/conversational structure and sophistication of ChatGPT or Gemini, with the technological capability of being agentic, i.e. being able to support users across their digital ecosystem (e.g. mobile phone, desktop, smart home, etc.) and "take actions" on their behalf (e.g. planning and booking trips, making reservations, etc.), whilst being increasingly personalised (e.g. maintaining an ongoing conversation with the same user and remembering facts about the user that was gathered from prior interactions or their digital footprint). Careful personalisation is crucial for these agents to be able to assist users effectively in their daily lives, as this increased sophistication could make users rely on them all the more, and for tasks that have serious real-world consequences. We expected the readers to be familiar with this development trend, but could have clarified it better with more examples.
>
> A typical use case of these assistants would be that the user asks the agent to assist them with planning an activity for themselves and/or others, like planning their daughter's birthday party. However, we noticed that this will require a continual awareness of user-specific safety risks, especially when users start to trust these agents to assist them effectively and in personalised ways (e.g. a user in a wheelchair, or who has a serious nut allergy, would likely expect their agent to take this fact about them into consideration in personalised recommendations). This form of risk has barely been discussed in prior work, and will become increasingly important once assistants start to function as a coherent "agent".
>
> To test how current leading models fare at such tasks, our multi-turn benchmark attempts to simulate a typical use case (across different conversational scenarios): i.e. the user engages in an ongoing dialogue with the same assistant, and then asks it whether it would recommend that they do a certain activity, assuming that the model remembers relevant facts about them or other relevant parties that they told the model about before. Our experiment demonstrates how the knowledge of user-specific safety critical constraints affects a dialogue assistant's behaviour in recommendation requests, and how this is further affected by the awareness of the desires of others.
>
> "more discussion of personalized LLM-based agents "
> The key contribution of our work is not proposing a novel approach to building personalised LLM agents, but a pipeline for evaluating personalised alignment, specifically the ability of models to account for person-specific safety risks. The suggested papers are more concerned with methods for enhancing personalisation, mainly with regards to accounting for user preferences, both of which are outside of our scope. Our reviewed work focused more specifically on prior considerations of person-specific safety alignment in the literature, and other approaches for multi-turn evaluation. The suggested papers could have been useful in our discussion section regarding potential ways of fixing the problems we encountered, although our work, by staying within the context window, showed that it was not a problem with memory but what a model attends to.

---

> > ### Comment · Reviewer_Cb54 · 2025-08-05
> >
> > Thank you to the authors for addressing my comments. I will maintain my positive scores.

---

### Official Review · Reviewer_EcYj · 2025-06-19

**Rating:** 2
**Confidence:** 4

**Summary:**

This paper introduces CURATe, a multi-turn benchmark for evaluating personalized alignment in LLM-based AI assistants, focusing on their ability to handle user-provided safety-critical contexts. The authors assess ten leading models across five scenarios (337 use cases each) and reveal systematic inconsistencies in maintaining user-specific considerations, with even top-rated "harmless" models making recommendations that are obviously harmful given user context. The work identifies key failure modes including inappropriate weighing of conflicting preferences, sycophancy, lack of attentiveness to critical user information, and inconsistent application of user-specific knowledge, proposing research directions for safer and more considerate AI assistants.

**Dataset Code Accessibility:**

Partly

**Dataset Code Comments:**

The submission is readily accessible and appears to be in its full and final form. The documentation is limited but probably sufficient. The dataset is hosted on a recognized platform (Github). There is no croissant metadata attached to the submission.

**Ethical Considerations:**

No, there are no or only very minor ethics concerns

**Limitations Weaknesses:**

* ISSUE A: The benchmark is a coherent measure; however, it is, I would argue, a coherent measure of something almost precisely the opposite of what the title, introduction and branding of the work present it to be.

The problem framing is roughly as follows; the user is presumed to have a bespoke safety-critical constraint; i.e., a soy allergy. There are also presumed to be other actors in the scenario with non-safety-critical constraints, i.e., they would like to eat tofu for dinner. The user then asks the LLM to adjudicate whether the user should violate his own safety-critical constraint. The implicit posture of the authors, then, is that "harmlessness", from an alignment standpoint, means that the LLM should never advise the user to do anything potentially unsafe, no matter how many other constraints are violated; something like "Asimov's first law of robotics by proxy, covering harms both emotional and physical". Setting aside the fact that this is not literally harmless (the LLM still causes harms to both the user and the proxies, albeit presumably lesser ones), I guess the implicit philosophy here would be roughly utilitarian, albeit without any theory to define or scope what the measure of utility would be.

But, and this is the key point, the promise of the paper is to benchmark *personalised alignment of conversational AI assistants*. In fact, this benchmark is doing just the opposite; establishing a fixed value structure and scoring AI assistants by how well they adhere to it. This conflates "alignment" with "paternalism" - assuming the model should protect users from themselves according to a fixed value hierarchy. A truly personalized system will respect a user's informed choice to prioritize social harmony over minor health risks, while still ensuring they understand the trade-offs.

* ISSUE B: The actual qualities measured by the benchmark are culturally informed and inherently subjective (will not necessarily generalize).

The authors write of a "much harder and under-explored challenge of being mindful of more pragmatic factors, effectively accounting for person-specific risks (e.g. irrational fears, severe allergies, physical constraints, trauma triggers) in how the agent treats and assists a given person. Depending on the sensitivities and personal facts a user expects the agent to know and remember about them, even seemingly benign or actively helpful utterances or recommendations can come across as rude or insensitive in certain contexts [1, 2], or put users at severe risk." An AI agent coming across as rude or insensitive cannot simply be asserted to be a 'severe risk' to a user. In fact, it's just as easy to assert the opposite; excessive sensitivity is a serious risk to users, putting them in sycophantic echo chambers that repeat their erroneous beliefs. In either case, the subject is not suitable for empirical, reproducible science of the kind this venue prioritizes.

* ISSUE C (minor): The authors argue that sycophancy means "prioritizing desires above safety". The textbook definition is "obsequious behavior toward someone important (in this case, a potential customer) in order to gain advantage." It would be helpful, in a work of this kind in particular, if the authors would use words according to their established and socially accepted meanings.

* ISSUE D (minor): The word "curate", which the benchmark acronym spells, doesn't link strongly to the topics of personalized aligment or AI safety and does link strongly to other unrelated AI research domains like data-centric ML. In other words, the acronym is unintentionally misleading.

* ISSUE E: The authors mention as a key finding "lack of attentiveness to critical user information within the context window" in their introduction and experiments section; context window attention issues have been extensively studied in the NLP literature, and the authors present this as if it's a novel finding from their benchmark rather than a well-known limitation being observed in a new setting.

https://arxiv.org/abs/2307.03172

https://arxiv.org/abs/2406.15981

https://arxiv.org/abs/2406.14673

https://arxiv.org/abs/2305.03731

* ISSUE F: The authors state, "strong reasoning capacities do not necessarily transfer to this kind of personalised thinking": however, Figure 2 shows that the o1 reasoning model scores higher on the benchmark than the all of the non-reasoning models (with the exception of the famously refusal-happy Llama 3 series, see https://arxiv.org/pdf/2406.11717 fig. 1).

**Strengths Contributions:**

The authors introduce a novel benchmark, CURATe. The benchmark is reasonably large, and what it measures is, to the best of my knowledge, not well covered in the existing literature. The benchmark is something of a hybrid between a safety benchmark, an NLP-style challenge of scaling counterfactuals, and a social science experiment, and seems to be a coherent measure. I found the results subjectively interesting to see.

---

> ### Author Response · Authors · 2025-08-01
> **Response to concerns - if users were supported in making an informed decision it was not counted as a Fail**
>
> We thank the reviewer for their thoughtful engagement with the paper and its content. We believe their main concerns are down to simple misunderstandings which we hopefully clarify below.
>
> Issue A It is not the case that "there are presumed to be other actors in the scenario with non-safety-critical constraints". We test the baseline in scenario 1, where no other actors are involved, and the model simply has to account for (i.e. mention at all) the user's safety-critical constraint. In that case, the recommendation is not serving any other actor and only putting the user at risk. We only added extra actors to test a secondary question, i.e if they do take the user's safety seriously, how seriously_ do they take it.
>
> You argue that a truly personalised system "will respect a user's informed choice to prioritize social harmony over minor health risks, while still ensuring they understand the trade-offs." By design, our benchmark is specifically designed to test for "severe" allergies or other serious constraints, not "minor health risks". All of our elements were created such that, if violated, they actually put the user in serious psychological or physical risk, or are actually physically impossible for them to do. We do agree with the second point, that recommending an activity is okay IFF the model is "still ensuring they understand the trade-offs", and have accounted for it in our experiment. That is what we were testing for in the ambiguous results, where the model recommended the activity whilst still mentioning the critical constraint in question, ensuring that the user is at least aware of the risk if they choose to proceed. To qualify as a pass, the model should not have even mentioned the constraint at all. Hence, in none of our passing rates were the safety-critical constraints even mentioned, while, interestingly, it typically does mention the preferences and desires of other actors. This poses a serious risk, as it does not help the user make an informed choice -- which, we argue, is crucial in potentially life-threatening situations. It is not a fixed value structure being imposed, as we are only interested in the model's ability to at least "notice" the constraint in question, and mention it in the recommendation. Rather, it is simply an illustration of how the knowledge of user-specific safety critical constraints affects a model's behaviour in recommendation requests, and how this is further affected by the awareness of the desires of others. The idea that it is good personalisation to prioritise the (non-essential) desires of one or more other actors over a serious risk to the user might be up for debate, but not to the extent that the agent completely forgets to even consider the safety of the user (i.e. failing to ensure they are aware of the risk to themselves). Perhaps you assumed fail rates could occur when the model "respects a user's informed choice, while ensuring they understand the trade-offs", which was not the case.
>
> Issue B: There's an important misunderstanding here. It is not the case at all that the qualities we test for are "inherently subjective", indeed we very carefully designed our experiment to only focus on things that would definitely make the given activity severely dangerous or even impossible to the specific user, given constraints they themselves mentioned as serious (e.g. things they're severely allergic to, seriously phobic of, traumatised by etc). None of our benchmark elements tests for simply being offended due to statements simply being "rude or insensitive". The quote was referring to prior work on forms of subjective harms. We expand on it by focusing on person-specific safety risks, a form of risk that is subjective only in that doing the activity is directly risky to a user with a given quality, not that it is potentially risky in any assumed social settings.
>
> Issue C: We appreciate the textbook definition of sycophancy. We were following the established trend of using sycophancy to refer to the noted "desire to please" in LLMs. Our point was that models are biased to be supportive of things people said they enjoy or like, as a part of this general "sycophantic" trend for model "agreeableness", which, we show, may lead models to fail to notice or mention conflicting safety-critical information. We understand that perhaps different terminology could have made this point clearer.
>
> Issue E: With "lack of attentiveness to critical user information within the context window" we were only making it clear that the lack of attentiveness was not due to memory constraints. This is not presented as a key finding, just a clarification of the source of the problem being one of attention and not memory. Our novel contribution is the kind of information being personalised to.
>
> Issue F: "refusal-happy Llama 3 series" is not relevant here, as refusals to respond, without mentioning the safety-critical fact, did not count as a pass rate but as ambiguous.

---

> > ### Comment · Reviewer_EcYj · 2025-08-03
> >
> > I thank the authors for taking the time to write the rebuttal; my opinion of the work is not changed, however. I am aware of the multiple experimental settings; since the mean pass rates appear to be generally above 90% for recently released models in Scenario 1 (which still includes random confounding questions), I presume it did not contribute strongly to the authors' claim in their abstract that they have "identified key failure modes". And the authors' use of words such as "severe" and "serious" are inherently subjective. For example, I happen to know parents of young children whose children have "severe" allergies who regularly (with medication) offer them the foods they are allergic to. In certain situations, this is actually part of the therapy for allergies. As for "traumatized", in some segments of society, the word is used as a sarcastic or comic signal.

---

> > > ### Author Response · Authors · 2025-08-04
> > >
> > > With failure modes, the main issues we identified is the (a) systematic decline in performance rates as more actors were introduced with non-critical desires, which should not affect a model's ability to *notice* critical user information that was shared, and (b) the relatively low performance on the user-specific safety reasoning task of most leading models that fare much better on more generic safety benchmarks. Over 90% was only achieved in 4/10 cases in the simplest conversational setup, but the addition of extra contextual information affected model attention significantly. Note that our experiment was not about simply recommending or not recommending the activity, but the ability of the model to at least *mention* (i.e. attend to) the fact that the activity is likely to conflict with a serious constraint that the user shared, so that the user is at least able to make an informed decision regarding their safety. Failures only occured when the model did not even mention the relevant 'critical' user constraint in their recommendation --- though we found that it typically did mention the relevant preference of the users' friends, which is a noteworthy bias.
> > >
> > > Random confounding questions should also not affect the models' abilities to attend to the relevant critical information; the reasoning behind this setup was to identify how irrelevant contextual information may affect models' abilities to attend to critical information even within the context window (e.g. relating to issues of attention and not memory), and our study setup was purposefully different to simple prompt-response pairs to better simulate ongoing conversations where handling longer contextual dependencies are important, as will be the likely use case in agentic assistants.
> > >
> > > As for the "subjective" nature of serious and severe, this is irrelevant, as the subjectivity comes from the user themselves, whom the agent is serving. If the user themself states their allergy or phobia as severe, then the model should take it seriously enough as to at least mention that the activity directly conflicts with the "serious" constraint they mentioned, because if indeed "severe" allergy means causing severe negative reactions, or even death (which is possible if not likely in many cases), a failure to mention that a food contains the allergen in question is very risky and could have "severe" consequences. This line of reasoning about how "traumatic" could be sarcastic is also irrelevant, as the user does not say "I am traumatised by..", but says "I have severe PTSD from...", which is not ambiguous in the way the reviewer suggests and should definitely be taken seriously. Regardless, the kinds of "sarcastic" or softer interpretations of "serious allergy" and "severe PTSD" are the exception not the rule; and for a benchmark that is primarily about assessing model safety, assuming those self-declarations by a user are probably subjective, sarcastic, and not even relevant safety information to consider in recommendations as the rule would be extremely irresponsible.
> > >
> > > Our work considers a class of risks that have been underexplored in the literature, about user-specific safety constraints, and how irrelevant contextual information affects models' abilities to mention those information in user-specific recommendations. Even if the reviewer believes our execution could be improved, we hope they can appreciate the value of raising it as a point of discussion in the community, as we are doing now. "Subjectivity" is a concept that is surely an important issue to consider in cases of personalisation, but the safest approach to this would surely be to stay aligned with the user's own opinion regarding which constraints count as serious, important, severe etc. or not, and take it seriously.

---

### Official Review · Reviewer_e2UE · 2025-07-02

**Rating:** 2
**Confidence:** 4

**Summary:**

This paper presents CURATe, a benchmark designed to evaluate the personalized alignment capabilities of LLM-based AI assistants, specifically their ability to retain and act on safety-critical user information across multi-turn conversations. The benchmark tests 10 models (from OpenAI, Meta, Google, Anthropic, and Mistral) across five interaction scenarios of varying complexity and four safety-critical user constraint types (e.g., severe allergy, PTSD, etc.). The authors reveal common model failures – including sycophancy, poor prioritization of safety, and context neglect – and show that standard alignment strategies (e.g., "helpful and harmless" prompting) are insufficient. They propose research directions involving self-reflection, user modeling, and dynamic risk assessment.

**Additional Feedback:**

See in weaknesses.

**Dataset Code Accessibility:**

Yes

**Dataset Code Comments:**

The code is accessible.

**Ethical Considerations:**

No, there are no or only very minor ethics concerns

**Final Justification:**

The response does not fully address all concerns. I appreciate the clarification on C and the partial clarification on A, but I am still unclear about B. In addition, the authors did not respond to some of the questions raised in my review. Therefore, I will maintain my original score.

**Limitations Weaknesses:**

**A. Limited Categories without theoretical support**
1. **Fixed scenario structure**: All conversations in the benchmark follow a rigid, 5-turn structure with predefined roles. While this design supports controlled benchmarking, the heavy reliance on templated, hand-crafted multi-turn interactions limits generalizability to more natural, less-structured dialogues. Results may not generalize to real user interactions or open-ended dialogue systems.

2. The study focuses on only four types of safety-critical constraints (e.g., allergies, trauma triggers). Although these are important, this excludes other forms of personalization. Moreover, the paper lacks a theoretical rationale for selecting the five specific interaction scenarios and four constraint types. Are these choices empirically or theoretically grounded? Are they representative of the space of risks LLMs may encounter?

**B. Limited Evaluation**
1. **Dependency on a Single Evaluator Model.**
All responses are evaluated using LLaMA 3.1 405B (Instruct). This raises several concerns: First, it is surprising that state-of-the-art models like GPT-4o were not used for evaluation. The authors do not provide a clear rationale, citation, or justification for selecting LLaMA 3.1 over more widely trusted or capable models. Why were GPT models excluded? Second, Relying on a single evaluator introduces the risk of bias, especially when that evaluator is also part of the benchmark. LLaMA 3.1 405B may favor responses generated by itself or LLaMA-style, or specific prompts. Although the paper reports high agreement between LLaMA and human judges for a subset of 100 samples, it is unclear which model generations these samples came from, especially since LLaMA performs best in the benchmark. It raises the possibility that using a different evaluator might yield significantly different results.

Suggestion: The authors should consider ensembling multiple evaluators or including more extensive human evaluations, particularly for ambiguous or borderline cases. Using evaluators that are architecturally distinct from the benchmarked models may reduce bias.

2. **Limited Evaluation Metrics.** The evaluation focuses solely on whether the assistant accounts for safety-critical constraints using a simple 3-point scale (fail, unclear, pass). However, no metrics are provided for generation quality, which is essential in dialogue systems. For example:

1). Does LLaMA 3 8B Instruct produce fluent and coherent responses? While it may follow instructions reasonably well, the paper does not evaluate whether the output is grammatically correct, redundant, or semantically meaningful.

2). The max generation length is 100 tokens. Will it result in truncated or repetitive outputs?

3). GPT-4o’s output is shown in Figure 8 as an example. However, no qualitative examples are provided for smaller, open-source models, such as Mixtral or LLaMA 8B.

4). The temperature setting is fixed at 0.7, which introduces non-determinism and may reduce reproducibility. It is also unclear whether the evaluator model used the same decoding configuration (e.g., temperature, max tokens) as the generation models.

**C. Prompt-based Limitations.**
Although guiding prompts significantly improve performance, this raises concerns about the brittleness of alignment mechanisms: models need to be told explicitly (e.g., "consider my safety constraints") to prioritize user constraints, suggesting a lack of inherent understanding – they may just follow cues.

**D. Concerns About Writing and Clarity.**

1. **Potential ChatGPT-Generated Sentences**: some phrasing – such as in lines 35-36 and 67-68, similar to ChatGPT format. e.g. “The ability of an AI assistant to maintain personalised alignment—consistently remembering and 36 appropriately acting upon relevant context and user-specific information—is crucial for safe and” may directly copy and paste from ChatGPT.

2. **Unclear Logic in Some Sections.** For example, the paragraph spanning lines 52–61 lacks clarity in distinguishing between general human preference alignment and personalized, safety-critical alignment. The motivation could be improved with a concrete example. For instance, for the sentence “Depending on the sensitivities and personal facts a user expects the agent to know and remember about them, even seemingly benign or actively helpful utterances or recommendations can come across as rude or insensitive in certain contexts [1, 2], or put users at severe risk.” would benefit from a detailed illustration. What specific kind of recommendation could seem helpful in general but severely harmful for a specific user?

**Questions and Clarifications:**
1. What does “the user and first person’s preferences/constraints” mean (line 193)?
2. What is the overall rating distribution (pass/fail/unclear) across all models and scenarios?
3. Inconsistencies in Figure 2: The paper states that performance "steadily declines" from Scenarios 1 to 4. However, Figure 2 contradicts this for some models. For example:
Claude 3.5 and OpenAI, Sc4 is slightly better than Sc3, and for Llama 3.1, Sc3 is slightly better than Sc2, while for other cases, Sc2 > Sc3 > Sc4. These deviations suggest the trend is not as uniform as claimed and should be discussed more explicitly.

**Strengths Contributions:**

1. The topic is timely, addressing a gap in LLM alignment research -- specifically, the lack of personalized, safety-aware evaluation benchmarks.

2. The benchmark is carefully constructed, featuring 337 use cases per scenario and incorporating controlled variables and ablations across five distinct scenarios.

3. The evaluation covers ten LLMs tested across increasingly complex conversational contexts. Results are clearly presented, with Figures 2 and 3 effectively illustrating model performance degradation as complexity increases, along with outcomes from various ablation studies.

4. The paper is overall well written, clearly structured, and easy to follow. The writing-related concerns are discussed under the weaknesses section.

---

> ### Author Response · Authors · 2025-08-01
> **Responses to concerns -- in defense of LLaMA**
>
> We thank the reviewer for their comments and thorough engagement with the paper's content. We respond to their concerns in turn below.
>
> A. Limited Categories without theoretical support
>
> "All conversations follow a rigid, 5-turn structure with predefined roles."
>
> This is not strictly true, as each of the 5 scenarios has a different number of conversation turns that were compared, involving different actors and information types. Variations on this structure were also compared in our ablation studies. It is true that all conversations involve the predefined roles of a user and an agent interacting with each other, but we believe this makes sense as it is the most typical use case. Whilst it would be interesting to test the problem out in more naturalistic settings (as we acknowledge ourselves in our future work section), we wanted to first boil the problem down to its simplest form: i.e. to nest critical user information somewhere within a small multi-turn conversation (at different places across ablations and scenarios) to see whether models are able to respond appropriately to it in the easiest case, and what factors contributed to success rates.
>
> "The study focuses on only four types of safety-critical constraints"
>
> These risks were chosen as they represent typical and common forms of personal risks, while at the same time satisfying the other constraint of being something that a normal recommendation algorithm would be likely to suggest. The authors first started to create benchmarks elements by hand, simulating different user scenarios in which personal sensitivities or risks were present. After a few dozen we started noticing that they largely fit into these four main categories. These were chosen for their commonality as well as severity, and the fact that all of the personal safety risks we could think of largely fell into these categories: i.e. physical risk (allergies and physical limitations like disabilities or chronic illnesses) and psychological risk (trauma triggers and phobias) in common or typical activity recommendation.
>
> B. Limited Evaluation
>
> "selecting LLaMA 3.1 over more widely trusted or capable models."
>
> Our justification for chosing LLama 405B is laid out in our appendix. Firstly, it is not true that other models are more capable. We chose it presicely because it consistently performed reliable assessments in our test experiments, where we manually evaluated outputs. If you look at current benchmarks LLaMA 405B is comparable and even outperforms models like GPT4o. It has the highest overall f-score balance at the time of writing. Its accuracy is also demonstrated in our human baseline comparison, achieving perfect and near-perfect accuracy against two human evaluators. The second reason was resource constraints. We could not afford to have GPT4o as an evaluator model. We initially tried the experiment with Claude 3.5., but found LLaMA to actually outperform it, and at a much lower cost. This also affects reproducibility, as the cost of running this experiment with our pipeline is thousands of pounds cheaper than it would be otherwise.
>
> "LLaMA 3.1 405B may favor responses generated by itself or LLaMA-style"
>
> None of the benchmark elements were generated by LLaMA. Anywhere that language models were involved in benchmark creation, we used either Claude 3.5 Sonnet (predominantly) and GPT4o (sometimes) to generate benchmark elements, as described in the appendix. In any case, we also explain why we believe that biases in this case is unlikely, as the benchmark elements were created individually (e.g. the specific information that was later nested randomly in different conversation turns), so no creator has full access to the conversation that was being tested.
>
> C. Limited Evaluation Metrics.
>
> Our experiment was not designed to test eloquence, as that was outside of our scope. We were only interested in a the pass rate on this specific task. Our pass/fail rates correspond only to clear recommendations for/against doing the activity, as any ambiguous responses are caught by the ambigious rating.
>
> "What recommendation could seem helpful in general but harmful for a specific user"
>
> Pretty much all of those in our benchmark, if helpfulness is understood as "a clear attempt to perform the task or answer the posed question". In our benchmark the models often try to be considerate towards other actors in group activity recs, putting the user at risk due to their unique constraints. That's what "user constraints and first person's preferences" refer to.

---

> > ### Comment · Reviewer_e2UE · 2025-08-08
> >
> > Thank you for the clarification. However, I am still confused about the “5-turn” setup (lines 151–163), where the authors state: “Each scenario follows this structure: xxx Turn 1, Turns 2–4, Turn 3, Turn 5.” In addition, the authors claim that “None of the benchmark elements were generated by LLaMA,” yet in lines 143–144, they list LLaMA 3.1 405B Instruct, LLaMA 3 70B Instruct, LLaMA 3 70B, and LLaMA 3 8B Instruct (Meta). I have also read other reviewers’ comments and rebuttals, and I agree with Reviewer EcYj as well as Reviewer Cb54 that the current presentation is somewhat difficult to follow. I will therefore maintain my original score.

---

### Decision · Program_Chairs · 2025-09-18

**Decision:**

Reject

**Comment:**

The paper introduces CURATe, a multi-turn benchmark designed to evaluate how LLM-based assistants are able to handle user-provided contexts in safety-critical contexts. The benchmark tests ten LLMs across five interaction scenarios and four safety-critical user constraints. The evaluation shows that all models systematically fail in maintaining use constraints and provide harmful recommendations. However, prompts that specifically target safety can reduce errors. The authors provide recommendations to advance safer and more reliable AI assistants through techniques like self-reflection, user modeling, and dynamic risk assessment.

Overall, the reviewers found the submission timely, relevant, and important (e2UE, Cb54), clearly structured, and easy to follow (e2UE). The benchmark is large (EcYj), well constructed, and covers a large number of LLMs and scenarios (e2UE, Cb54), while the results are clearly presented (e2UE), together with ablation studies (e2UE).

Several limitations have been also mentioned: 1) lack on methodological clarity with regard to how the conversations were taking place (e2UE), 2) the use of a single evaluator (e2UE), 3) limited evaluation metrics (e2UE) that are likely 4) to not generalize (EcYj) , 5) lack of comparison with related benchmarks (Cb54), and 6) missing important literature that could affect the novelty of the method presented (EcYj).

While the rebuttal and the discussions clarified some of the issues regarding the evaluation metrics and the setup for the conversations, there are still major weaknesses of the submission that may not be easily addressed without a thorough revision. As such, while the submission has promising aspects, I tend to a reject recommendation.